# Effects of La on Thermal Stability, Phase Formation and Magnetic Properties of Fe–Co–Ni–Si–B–La High Entropy Alloys

**Jiaming Li, Jianliang Zuo and Hongya Yu ***

School of Materials Science and Engineering, South China University of Technology, Guangzhou 510640, China; johnlee1215@163.com (J.L.); zuojianliang@126.com (J.Z.)
* Correspondence: yuhongya@scut.edu.cn

**Abstract:** The microstructure, phase formation, thermal stability and soft magnetic properties of melt-spun high entropy alloys (HEAs) $Fe_{27}Co_{27}Ni_{27}Si_{10-x}B_9La_x$ with various La substitutions for Si ($x$ = 0, 0.2, 0.4, 0.6, 0.8, and 1) were investigated in this work. The $Fe_{27}Co_{27}Ni_{27}Si_{10-x}B_9La_{0.6}$ alloy shows superior soft magnetic properties with low coercivity $H_c$ of ~7.1 A/m and high saturation magnetization $B_s$ of 1.07 T. The content of La has an important effect on the primary crystallization temperature ($T_{x1}$) and the secondary crystallization temperature ($T_{x2}$) of the alloys. After annealing at relatively low temperature, the saturation magnetization of the alloy increases and the microstructure with a small amount of body-centered cubic (BCC) phase embedded in amorphous matrix is observed. Increasing the annealing temperature reduces the magnetization due to the transformation of BCC phase into face-centered cubic (FCC) phase.

**Keywords:** magnetic materials; high entropy alloys; thermal stability; phase transformation

## 1. Introduction

High entropy alloys (HEAs), defined as the alloys consisting of at least five principal elements without obvious base element, have been proposed by Cantor et al. [1] and Yeh et al. [2] in 2004, independently. Up to now, there are two commonly used definitions of HEAs. One is the composition-based concept, i.e., the alloys composed of five or more principal elements in equal or near equal molar ratio between 5 atom percent (at.%) and 35 atom percent (at.%). The other definition is based on total configurational molar entropy ($S_{mix}$). The alloys with $S_{mix}$ < 1 R, 1 R < $S_{mix}$ < 1.5 R, and $S_{mix}$ >1.5 R, where R is the gas constant, are defined as low entropy alloys, medium entropy alloys, and high entropy alloys, respectively [3,4]. As a new type of alloys with unique properties of high strength, hardness, corrosion resistance, abrasion resistance and high fatigue resistance, HEAs have received extensive attention. Instead of forming a complex crystal structure, HEAs usually tend to form a solid solution with a face-centered cubic (FCC) or body-centered cubic (BCC) structure, or a mixture thereof [5,6], although a hexagonal close-packed (HCP) structure may be found in a few of HEAs [7].

Studying the compositions, microstructure and their fundamental properties to establish a fundamental database is currently the most essential work for HEAs [8,9]. Up to now, a series of HEAs have been prepared, including Fe-based, Co-based [10], Fe–Co–Ni-based [11] and rare earth-based high-entropy alloys [12,13]. However, most of the previous work focused on their mechanical properties and microstructure [9,14], and their physical properties have not been fully investigated. As Fe, Co and Ni are common constituent elements used in HEAs [15], it is very interesting to explore the magnetic properties of the HEAs. As we know, the soft magnetic materials are developing towards low coercivity ($H_c$) and high magnetization ($M_s$), which are essential for promoting the energy conservation efficiency and miniaturization of the electromagnetic device. However, some existing reports on the high-entropy soft magnetic alloys indicate that the saturation induction $B_s$

of the HEAs is still low, typically less than 1 T and their crystallization temperature ($T_x$) is also less than 670 K [16–19], which are both less than what we expected.

On the other hand, the rare earth elements (RE) have been frequently employed in the soft magnetic alloys [20,21], and the results showed that the addition of RE elements such as Gd and Tb can increase the curie temperature [20], and modify the crystallization temperature of the alloy. The addition of RE can also decrease the magnetic permeability. However, the influence of trace rare earth elements on HEAs has rarely been studied. In this work, La is selected to substitute Si for improving the performance of Fe–Co–Ni–Si–B HEAs. La exhibits low solubility with Fe, Co, and Ni elements, and it may play a role of micro-alloying. The thermodynamic properties, glass-forming ability (GFA) and magnetic properties of Fe–Co–Ni–Si–B–La HEAs are studied in detail.

## 2. Experimental

The alloy ingots of $Fe_{27}Co_{27}Ni_{27}Si_{10-x}B_9La_x$ with x = 0, 0.2, 0.4, 0.6, 0.8, and 1 (atomic ratio), denoted as $La_0$, $La_{0.2}$, $La_{0.4}$, $La_{0.6}$, $La_{0.8}$, and $La_1$, respectively, were prepared by arc-melting pure Fe (99.5 wt.%), Co (99.9 wt.%), Ni (99.96 wt.%), La (99.9 wt.%) metals, FeB (with Fe 83.78 wt.% and B 16.22 wt.%) and Si (99.99 wt.%) crystals under argon atmosphere. The ingots were melted 5 times to ensure chemical homogeneity. The ribbons with width of ~1.2 mm and thickness of ~0.025 mm were prepared by single-roller melt spinning method with the wheel speeds of 45 m/s. The phase structures of the alloys were characterized by X-ray diffraction (XRD) with Cu K$\alpha$ radiation. Thermal stability was studied by differential scanning calorimetry (DSC) at a heating rate of 10 K/min and under argon atmosphere. The saturation magnetization ($B_s$) of ribbons were measured under an applied field of 250 kA/m with a vibrating sample magnetometer (VSM). The coercive force ($H_c$) was measured with a MATS-2010SD hysteresis curve (DC) test system using ribbons about 50 mm in length.

## 3. Results and Discussion

It is reported that HEAs trend to form simple fcc and/or bcc solid solution structure or metallic glass. Figure 1 shows the XRD pattern of arc melt $Fe_{27}Co_{27}Ni_{27}Si_{10-x}B_9La_x$ (x = 0, 0.6, and 1) ingots. In all the ingots, the fcc phase, $(FeCoNi)_2B$ and $Ni_{31}Si_{12}$ phases were detected [18,19].

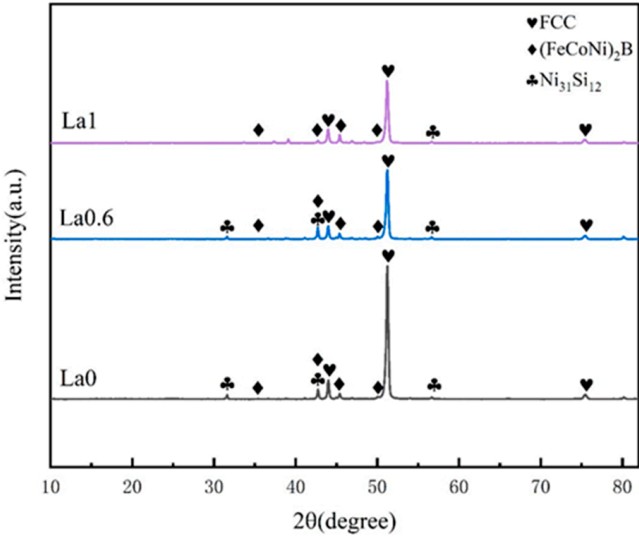

**Figure 1.** The XRD pattern of as-cast $Fe_{27}Co_{27}Ni_{27}Si_{10-x}B_9La_x$ (x = 0, 0.6, 1) ingots.

According to prior research [22], there are many phases in high entropy alloys, including solid solutions, intermetallic compounds, and amorphous phases. The phase evolution in HEAs can be predicted mainly by three parameters, namely atomic size difference

($\Delta$), mixing enthalpy ($\Delta H_{mix}$), mixing entropy ($\Delta S_{mix}$) and valence electron concentration (*VEC*) [23,24]. The $\Delta$, $\Delta H_{mix}$, $\Delta S_{mix}$, and *VEC* are defined as:

$$\delta = 100 \sqrt{\sum_{i=1}^{N} c_i \left(1 - \frac{r_i}{\bar{r}}\right)^2} \tag{1}$$

$$\Delta H_{mix} = \sum_{i=1,i\neq j}^{N} \Omega_{ij} c_i c_j \tag{2}$$

$$\Delta S_{mix} = -R \sum_{i=1}^{N} c_i \ln c_i \tag{3}$$

$$VEC = \sum_{i=1}^{N} c_i (VEC)_i \tag{4}$$

where $N$ is the number of the components in HEAs, $R$ is gas constant, $c_i$ is the atomic fraction of $i$-th component, and $\bar{r}$ is the average atomic radius. $r_i$ is the atomic radius, which can be obtained from References [15,24]. *VEC*, $\Delta H_{mix}$, and $\Delta S_{mix}$ between atomic pairs also be obtained in References [15,24]. The values of $\Delta$, $\Delta H_{mix}$, $\Delta S_{mix}$, and *VEC* for $Fe_{27}Co_{27}Ni_{27}Si_{10-x}B_9La_x$ alloys are summarized in Table 1. It is clear that all *VEC* values are near 8.0. Guo et al. [25] pointed out that fcc phase forms in the alloy with $VEC \geq 8.0$, bcc phase forms at $VEC \leq 6.87$, and a mixture of fcc and bcc phases at $6.87 \leq VEC \leq 8.0$. Hence, these $Fe_{27}Co_{27}Ni_{27}Si_{10-x}B_9La_x$ alloys trend to form fcc solid solution and intermetallic compounds.

**Table 1.** The atomic radius difference ($\Delta$), valence electron concentration (*VEC*), mixing enthalpy ($\Delta H_{mix}$), mixing entropy ($\Delta S_{mix}$) and structure of the $Fe_{27}Co_{27}Ni_{27}Si_{10-x}B_9La_x$ (x = 0, 0.6, 1) alloy systems (atomic percent).

| Sample | $\Delta$ (%) | $\Delta S$ (kJ/mol) | $\Delta H$ (kJ/mol) | *VEC* | Structure |
|---|---|---|---|---|---|
| $Fe_{27}Co_{27}Ni_{27}Si_{10}B_9$ | 10.2 | 12.615 | −20.42 | 7.96 | FCC + IM |
| $Fe_{27}Co_{27}Ni_{27}Si_9B_9La_{0.6}$ | 11.06 | 12.722 | −19.31 | 7.954 | FCC + IM |
| $Fe_{27}Co_{27}Ni_{27}Si_9B_9La_1$ | 11.58 | 12.804 | −18.66 | 7.95 | FCC + IM |

　　　Figure 2a shows the XRD patterns of as-spun $Fe_{27}Co_{27}Ni_{27}Si_{10-x}B_9La_x$ alloys. Only a broad diffraction peak appears at near 45° without any detectable crystalline peaks for all alloys, indicating fully amorphous structure. Figure 2b shows the DSC curves of the as-spun $Fe_{27}Co_{27}Ni_{27}Si_{10-x}B_9La_x$ ribbons. All curves have two distinct exothermic peaks and one endothermic peak, giving two-stage crystallization and melting processes. The glass transition temperature ($T_g$), phase transition temperature ($T_p$) [16], liquidus temperature ($T_l$), primary crystallization temperature ($T_{x1}$), and secondary crystallization temperature ($T_{x2}$) are marked by arrows. As shown in Figure 2b, The $T_g$ of amorphous ribbons range from 642 to 694 K. The $T_{x1}$ and $T_{x2}$ for the alloys with different La contents are in the region of 707–743 K and 802–839 K, respectively. The $T_{x1}$ initially increases from 722 to 743 K with increasing La content from 0 to 0.2 at.%, and then decreases to 707 K with increasing La to 1 at.%. The largest $T_{x1}$ of 743 K is obtained in the alloy with 0.2 at.% La substitution. Similarly, $T_{x2}$ increase from 802 to 839 K with further increase of La. At 1 at.% La substitution, $T_{x2}$ reaches the largest value of about 839 K. The value of $\Delta T_x$ (= $T_x - T_g$) of these alloys are in the region of 43–65 K, and it becomes large as x increases up to 1, which indicates that less than 1 at.% La substitution is beneficial to forming amorphous structure and hindering crystallization process [26]. The large $\Delta T_x$ up to 65 K for $Fe_{27}Co_{27}Ni_{27}Si_9B_9La_1$ alloy shows good thermal stability of the supercooled liquid. In addition, $T_l$ of alloys increases from 1318 to 1439 K as x increase from 0 to 0.6 at.%, then decreases to 1324 K with x increases to 1 at.%. The $T_g$, $T_x$, $T_l$, $\Delta T_x$, reduced glass

transition temperature $T_{rg}$ (= $T_g/T_l$) [27], and S (= $\Delta T_x/(T_l - T_g)$ [28] are listed in Table 2. The *S* values and $T_{rg}$ values exhibit good correlation with $\Delta T_x$, and the largest *S* value of 0.096 is obtained at x = 1.

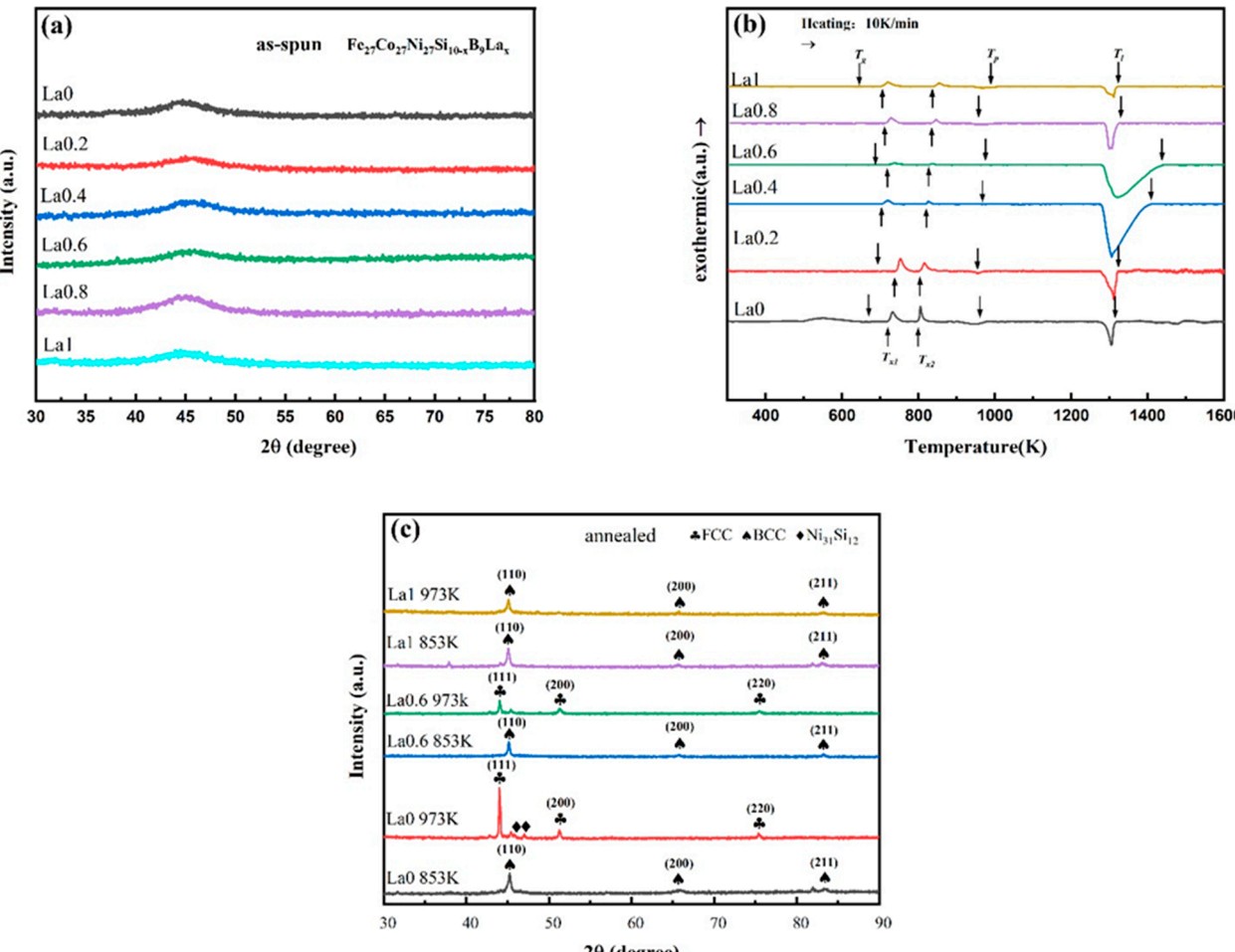

**Figure 2.** (**a**) XRD patterns of as-spun $Fe_{27}Co_{27}Ni_{27}Si_{10-x}B_9La_x$ (x = 0 to 1) alloys, (**b**) DSC curve of as-spun $Fe_{27}Co_{27}Ni_{27}Si_{10-x}B_9La_x$ alloys, and (**c**) XRD patterns of amorphous ribbons annealed for 5 min at 853 and 973 K.

**Table 2.** Thermal parameters of $Fe_{27}Co_{27}Ni_{27}Si_{10-x}B_9La_x$ amorphous ribbons.

| Composition (at.%) | $T_g$ (K) | $T_{x1}$ (K) | $T_{x2}$ (K) | $\Delta T_x$ (K) | $T_l$ (K) | $T_{rg}$ | S |
|---|---|---|---|---|---|---|---|
| $Fe_{27}Co_{27}Ni_{27}Si_{10}B_9$ | 667 | 722 | 802 | 55 | 1318 | 0.506 | 0.084 |
| $Fe_{27}Co_{27}Ni_{27}Si_{9.8}B_9La_{0.2}$ | 694 | 743 | 806 | 49 | 1324 | 0.524 | 0.078 |
| $Fe_{27}Co_{27}Ni_{27}Si_{9.8}B_9La_{0.4}$ | - | 710 | 823 | - | 1407 | - | - |
| $Fe_{27}Co_{27}Ni_{27}Si_{9.8}B_9La_{0.6}$ | 680 | 723 | 830 | 43 | 1439 | 0.473 | 0.057 |
| $Fe_{27}Co_{27}Ni_{27}Si_{9.8}B_9La_{0.8}$ | - | 713 | 832 | - | 1327 | - | - |
| $Fe_{27}Co_{27}Ni_{27}Si_{9.8}B_9La_1$ | 642 | 707 | 839 | 65 | 1321 | 0.486 | 0.096 |

To further study the crystallization behavior of $Fe_{27}Co_{27}Ni_{27}Si_{10-x}B_9La_x$ ribbons with different La contents, the $Fe_{27}Co_{27}Ni_{27}Si_{10-x}B_9La_x$ (x = 0, 0.6, 1) ribbons were annealed at different temperatures. Figure 2c shows the XRD patterns of $Fe_{27}Co_{27}Ni_{27}Si_{10-x}B_9La_x$ (x = 0, 0.6, 1) alloys after annealing at 853 and 973 K (above $T_p$) for 5 min. After annealing at 853 K, between $T_{x1}$ and $T_{x2}$, a bcc phase precipitates in the amorphous matrix. With the annealing temperature increased to 973 K, the bcc phase disappeared and fcc crystals formed. The transformation of bcc phase to fcc phase can also be observed in Fe–Co–Ni–Si–B HEAs at high temperature [29]. Combined with the DSC analysis results, the first exothermic peak is due to the precipitation of bcc phase and the second peak originates

from bcc phase and $Ni_{31}Si_{12}$ phases, and $T_p$ represents the transformation of bcc phase to fcc phase. For the alloy without La-substitution, after annealing at 973 K, a small amount of intermetallic compounds was detected, indexed as $Ni_{31}Si_{12}$ phase. Previous research has demonstrated that the Ni element is easy to segregate from Fe-rich bcc phases, resulting in the formation of fcc phase, and the over-saturated Si in Ni may form the Ni-Si intermetallic compounds [29,30]. However, after La addition, no $Ni_{31}Si_{12}$ phase was detected in annealed samples. Previous study [29] also showed that in Fe–Co–Ni–Si–B HEAs, $Ni_{31}Si_{12}$ phase could exist in high temperature. Thus, in the present alloys, the addition of La can suppress the formation of $Ni_{31}Si_{12}$ phase. Based on above discussion, the phase transition in $Fe_{27}Co_{27}Ni_{27}Si_{10-x}B_9La_x$ amorphous alloys after annealing occurs through the process of amorphous → amorphous' + bcc phase + $Ni_{31}Si_{12}$ → fcc phase.

Figure 3a shows the magnetic hysteresis loops (*M–H* curves) of as-spun $Fe_{27}Co_{27}Ni_{27}Si_{10-x}B_9La_x$ (x = 0 to 1) alloys. All alloys show soft magnetic behavior. The saturation magnetization $M_s$ of these alloys increases from 0.86 T to 1.01 T as x increases from 0 to 0.4, and then decreases to 0.88 T with x increasing to 1. Figure 3b shows the *M–H* curves of the $Fe_{27}Co_{27}Ni_{27}Si_{10-x}B_9La_x$ (x = 0, 0.6, 1) alloys after annealing at 573 K (below glass transition temperature) and 703 K (below crystallization temperature) for 5 min. The saturation magnetization $M_s$ increases with the increasing annealing temperature. The $M_s$ values of the alloys with x = 0, 0.6 and 1 annealed at 703 K is about 0.96 T, 0.99 T and 0.97 T, respectively. The coercivity $H_c$ values were measured as 10.3, 7.1, and 8.5 A/m at 573 K, respectively. As the heat treatment temperature increased to 703 K, the coercivity values were obtained as 18.4, 22.6 and 12.6 A/m for $Fe_{27}Co_{27}Ni_{27}Si_{10-x}B_9La_x$ (x = 0, 0.6, and 1) alloys.

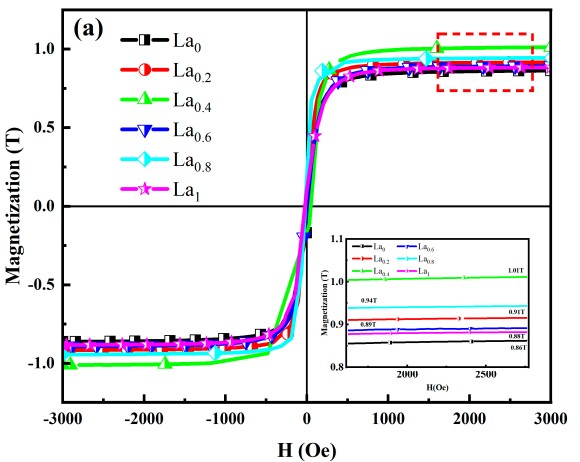 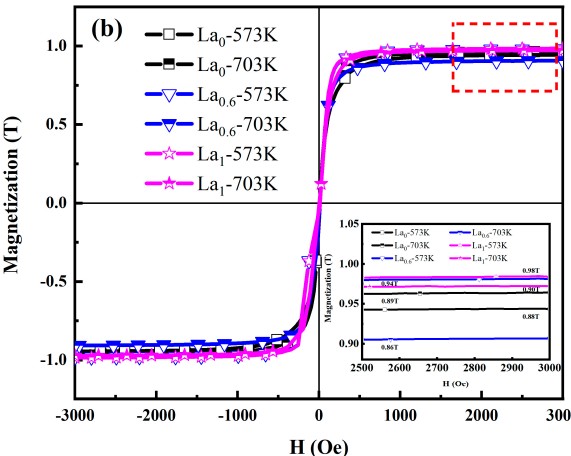

**Figure 3.** (**a**) The *M–H* curves of as-spun $Fe_{27}Co_{27}Ni_{27}Si_{10-x}B_9La_x$ (x = 0 to 1) alloys, and (**b**) the *M–H* curves of $Fe_{27}Co_{27}Ni_{27}Si_{10-x}B_9La_x$ (x = 0, 0.6, 1) alloys after heat treatment at different temperatures for 5 min.

As the annealing temperature rises above the crystallization temperature, the solid solution phase or other phases precipitate in the alloy. It is important to confirm the influence of the precipitation of the solid solution phase on the soft magnetic properties of HEAs. The magnetization curves of the annealing $Fe_{27}Co_{27}Ni_{27}Si_{10-x}B_9La_x$ (x = 0, 0.6, and 1) ribbons are shown in Figure 4. After annealing at 853 K, the saturation magnetization of the HEAs ribbons is increased. The $M_s$ values of the alloys with x = 0, 0.6 and 1 annealed at 853 K is about 1.05 T, 1.06 T and 1.07 T, respectively. With increasing annealing temperature to 973 K, $M_s$ and $H_c$ decrease simultaneously. The $M_s$ values of the alloys with x = 0, 0.6 and 1 is about 0.93 T, 0.89 T and 1.0 T, respectively. After annealing at higher temperature, the values of $M_s$ increased, but the coercivity was deteriorated dramatically. As shown in Figure 5, the coercivity $H_c$, after crystallization annealing is greatly increased. This phenomenon may be related to the fine grains precipitated in the amorphous matrix. Small crystal grains hinder the movement of magnetic domains and play a pinning role. According to the current experimental data, the addition of La element can increase the recrystallization temperature, so an appropriate amount of La can reduce the effect of heat treatment on

the reduction of saturation magnetization (sample $La_1$ have highest $B_s$ after annealing at 973 K). At the same time, the coercivity of the alloys with La element after annealing is relatively small.

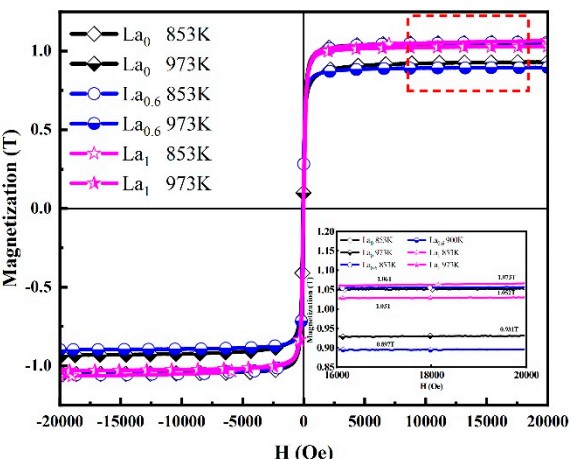

**Figure 4.** The *M–H* curves of $Fe_{27}Co_{27}Ni_{27}Si_{10-x}B_9La_x$ (*x* = 0, 0.6, 1) alloys annealed above the crystallization temperature.

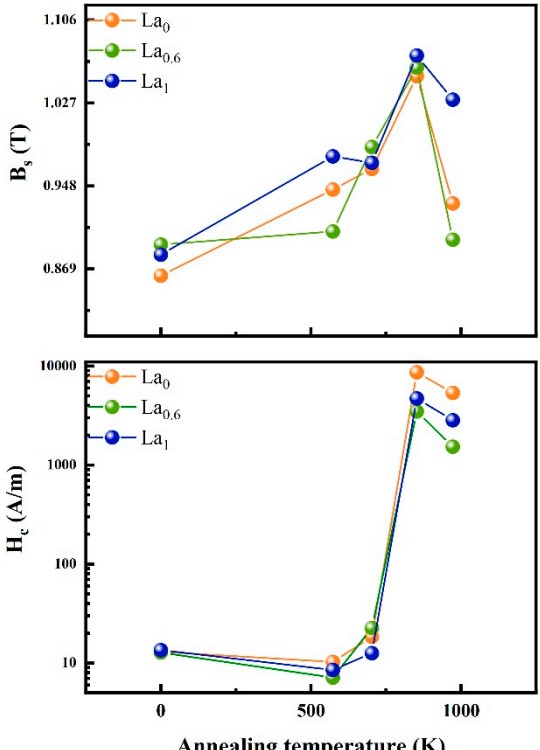

**Figure 5.** Changes in saturation magnetization ($B_s$), coercivity ($H_c$) with different annealing temperature for $Fe_{27}Co_{27}Ni_{27}Si_{10-x}B_9La_x$ (*x* = 0, 0.6, 1) alloys.

The current results thus show that La element substitution for Si has a great influence on the crystallization temperature of the alloy, and it is interesting that La can inhibit the formation of intermetallic compounds. Furthermore, under the same heat treatment conditions, the saturation magnetization of the strip with a certain La content is higher, while the coercivity is relatively lower.

## 4. Conclusions

A new type of soft magnetic $Fe_{27}Co_{27}Ni_{27}Si_{10-x}B_9La_x$ HEAs were developed in this work. The effects of La on the phase stability, amorphous forming ability and magnetic properties of Fe–Co–Ni–Si–B HE-MGs were investigated. It was found that the soft magnetic properties of $Fe_{27}Co_{27}Ni_{27}Si_{10-x}B_9La_x$ HEAs can be effectively tailored by adjusting their phase structure by annealing treatment. These alloys exhibit a low $H_c$ and a high $B_s$, in which the values are less than 25 A/m and higher than 1.0 T, respectively. The La content has an important effect on the values of $T_{x1}$ and $T_{x2}$ of the alloys. By increasing the annealing temperature, these alloys precipitated the BCC phase at the first crystallization temperature and transformed into the FCC phase at the phase transition temperature. In additionally, La can inhibit the formation of intermetallic compounds at high temperatures. This work suggests that an optimized annealing temperature is required to obtain good combination of the soft magnetic properties for HEAS.

**Author Contributions:** Conceptualization, J.L. and J.Z.; methodology and data curation, J.L.; writing-original draft preparation, J.L.; supervision, H.Y. All authors have read and agreed to the published version of the manuscript.

**Funding:** This work was supported by Guangdong Provincial Natural Science Foundation of China (No. 2021A1515010642).

**Acknowledgments:** All individuals included in this section have consented to the acknowledgement.

**Conflicts of Interest:** The authors declare no conflict of interest.

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
