# Peer review of "Effects of La on Thermal Stability, Phase Formation and Magnetic Properties of Fe–Co–Ni–Si–B–La High Entropy Alloys"

_metals, doi:10.3390/met11121907_

Round 1

Reviewer 1 Report

The study focuses on a new type of soft magnetic high entropy alloys, namely Fe27Co27Ni27Si10-xB9Lax, where La is substituting Si for improving the performance of the alloy. The effect of La on the phase stability, amorphous forming ability and magnetic properties of Fe-Co-Ni-Si-B HE-MGs is investigated. The thermodynamic properties, glass-forming ability (GFA) and magnetic properties of Fe-Co-Ni-Si-B-La HEAs are thoroughly studied. It was found that La content has important effect on the values of the crystallization temperatures of the amorphous alloys. It is also found that the soft magnetic properties of Fe27Co27Ni27Si10-xB9Lax HEAs can be effectively tailored by adjusting their phase structure by appropriate annealing. The work suggests that an optimized annealing temperature is required to obtain good combination of the soft magnetic properties for this type of HEAs.

I find the results reliable and useful and on this base I think that the work can be published.

Some notes to be taken into account:

1) the size of the quenching disk is not given and therefore the determined thickness of the ribbon (0.25 mm) seems large at this surface velocity of the disk. Could you check the values?

2) from the DSC curves  it is difficult to see the glass transition transition. May be some inset??

Author Response

  1. I have check the thickness of the ribbon, the correct value is 0.025 mm. This is a typo, thank you very much for your careful correction.
  2. the detail of the DSC curves please see the attachment.

Reviewer 2 Report

Authors of this article have studied the influence of doped La on the crystallographic structure, phase transition, and magnetic properties of melt-spun Fe27Co27Ni27Si10-xB9Lax (x = 0 -1) HEAs. The study if of high originality/novelty. However, both the quality of presentation and the scientific soundness are low. The results are not clearly presented. In addition, extensive editing of English language and style is required.

  • Line 22: “without no” may need to be replaced by “without” or “with no”.
  • Lines 32 – 34: The authors claimed that “Instead of forming complex crystal structure, HEAs usually tend to form bulk metallic glass (BMG) structure or intermetallic compounds (IM) with face-centered cubic (FCC), body-centered cubic (BCC) or their mixture structure [5, 6], although hexagonal close-packed (HCP) structure may be found in a few of HEAs [7].”

This above description is not correct. This is because most HEAs are neither BMG nor IM. Instead, they are mainly fcc and/or bcc-structured solid solutions.

  • Lines 67 - 68: Bs may have to be replaced by Ms. This is because Bs is often used to represent saturation induction.
  • Page 3: The authors calculated four parameters but only discussed one of the four parameters - VEC. If the authors did not discuss the other three parameters, then why did they calculate and list them?
  • Line 94: at near 45o. Here o should be a superscript.
  • Lines 113 – 114: the authors claimed that “The S values and Trg values exhibit good correlation with ΔTx”. This is not true, as judged from the values listed in Table 2. Data listed in Table 2 suggest that S correlates well with ΔTx rather than Trg. The authors should discuss the reasons.
  • Pages 5 and 6: the authors claimed M-H cures in the caption of figures 3 and 4 but used B(T) for the vertical axis in figures 3 and 4. This will cause confusing. B(T) should be better represented by M(T). Bs should also be replaced by Ms.
  • More discussions are needed to address the influence of annealing temperature and La on saturation magnetization and coercivity shown in Fig.5.
  • The authors mentioned that they measured the maximum permeability in Experimental section. However, they did not present and discuss the permeability data in their manuscript.
  • It is impossible to read the three insets in Figs.3 and 4.

Author Response

  1. Line 22: the "without no" have been repalced to "without"
  2. Line 32-34: The description you mentioned is correct, thanks for correcting
  3. Line 67-68: Bs have been replaced by Ms
  4. Page 3: The other three parameters are important of judge the formation of phase, but no clear which parameters can predict the phase correctly, so I calculate all the four parameters, then figure out the more important parameter for the alloys,and maybe the reader can know the important of  the parameters well if them be presented together.
  5. Line 94: the mistake has been corrected
  6. Line 113-114: Thanks for the suggestion, I have modified the sentence
  7. Page 5 and 6: the picture and text have been modified
  8. More discussion of Fig 5 have been added
  9. The problem have been modified
  10. Fig. 3 and Fig. 4 have been replaced
  11. Please see the attachment. Attached is the revised paper, thank you very much for taking the time to read.

Reviewer 3 Report

This is a rather interesting little experimental work aimed at studying how the presence of lanthanides (in this case, lanthanum) affects the properties (primarily magnetic) of high-entropy alloys. I think that the results obtained will be of interest to readers. However, in my opinion, some text corrections are required before posting:

1. Lines 27-28. Error. The boundary of high-entropy alloys is 1.5R. If we use the 1.5R boundary, none of the alloys the authors work with is high-entropy. The lower limit of medium-entropy alloys is also not 0.69R, but 1.0R.
2. The composition of the investigated system is not fully justified. Lanthanum replaces silicon, but why is silicon there? Why is there boron? The presence of these elements will obviously lead to the formation of compounds that violate the homogeneous structure of the high-entropy alloy - one can be sure of this without resorting to calculations according to formulas 1-4, which in this case do not make sense at all.
3. Lines 141, 153, 159. There is a dot after "Figure". What for?

Author Response

  1. Thanks for your correction. I rechecked the definition of high entropy alloys. The boundary is 1R and 1.5R exactly. I recalculated the alloy's mixing entropy, which is higher than 1.5R, the definition of compound high-entropy alloy.
  2. The original motivation for the existence of silicon and boron is to improve the ability to form amorphous, which is in line with the purpose of preparing amorphous ribbons.
  3. I have corrected the error in Line 141, 153, 159.
  4. Please see the attachment. The revised article is in the attachment, thank you for taking the time to read it.

Round 2

Reviewer 2 Report

English of this article still needs to be improved. 

Author Response

1.Thank you very much for taking the time to read this article from your busy schedule. I modified some sentences in the article to make the article more readable.
